# μ-Theraphotoxin Pn3a inhibition of Ca$_V$3.3 channels reveals a novel isoform-selective drug binding site

**Jeffrey R McArthur[1]\*, Jierong Wen[2], Andrew Hung[2], Rocio K Finol-Urdaneta[1], David J Adams[1]**

[1]Illawarra Health and Medical Research Institute, University of Wollongong, Wollongong, Australia; [2]School of Science, RMIT University, Melbourne, Australia

**Abstract** Low voltage-activated calcium currents are mediated by T-type calcium channels Ca$_V$3.1, Ca$_V$3.2, and Ca$_V$3.3, which modulate a variety of physiological processes including sleep, cardiac pace-making, pain, and epilepsy. Ca$_V$3 isoforms' biophysical properties, overlapping expression, and lack of subtype-selective pharmacology hinder the determination of their specific physiological roles in health and disease. We have identified μ-theraphotoxin Pn3a as the first subtype-selective spider venom peptide inhibitor of Ca$_V$3.3, with >100-fold lower potency against the other T-type isoforms. Pn3a modifies Ca$_V$3.3 gating through a depolarizing shift in the voltage dependence of activation thus decreasing Ca$_V$3.3-mediated currents in the normal range of activation potentials. Paddle chimeras of K$_V$1.7 channels bearing voltage sensor sequences from all four Ca$_V$3.3 domains revealed preferential binding of Pn3a to the S3-S4 region of domain II (Ca$_V$3.3$^{DII}$). This novel T-type channel pharmacological site was explored through computational docking simulations of Pn3a, site-directed mutagenesis, and full domain II swaps between Ca$_V$3 channels highlighting it as a subtype-specific pharmacophore. This research expands our understanding of T-type calcium channel pharmacology and supports the suitability of Pn3a as a molecular tool in the study of the physiological roles of Ca$_V$3.3 channels.

**\*For correspondence:** jeffreym@uow.edu.au

## Editor's evaluation

Low voltage activated T-type calcium channels (CaV3.1-CaV3.3) are important to several physiological processes, but the functions of individual isoforms are difficult to distinguish pharmacologically. This study reports a spider toxin, Pn3a, which exhibits 100-fold selectivity for inhibiting CaV3.3 over CaV3.1 and CaV3.2 isoforms. This specificity is conferred by a specific interaction site on the Cav3.3 domain II voltage sensor.

## Introduction

Voltage-gated calcium (Ca$_V$) channels are activated by membrane depolarization and are involved in many physiological processes including contraction, secretion, neurotransmitter release, and gene expression (*Catterall et al., 2005*). In contrast to high voltage-activated (HVA) Ca$^{2+}$ currents mediated by L-, N-, P/Q-, and R-type calcium channels that require large depolarizations, low voltage-activated (LVA) currents mediated by T-type Ca$^{2+}$ channels are activated by small membrane depolarization and display distinctively faster activation and inactivation kinetics. T-type channel activation near resting membrane potentials generates low-threshold Ca$^{2+}$ spikes responsible for the conspicuous burst firing and low-frequency oscillatory discharges observed in thalamic, olivary, and cerebellar neurons (*Park et al., 2010*; *Dreyfus et al., 2010*; *Molineux et al., 2006*; *Lee et al., 2014*). Thus, the biophysical

properties of the T-type channels make them important regulators of cardiac and neuronal excitability (*Perez-Reyes, 2003*; *Huguenard, 1996*; *Coulter et al., 1989*) and therefore key pharmacological targets for the treatment of neurological and psychiatric disorders (*Zamponi, 2016*; *Maksemous et al., 2022*).

T-type calcium channels display small single-channel conductance (T stands for transient or tiny) (*Perez-Reyes et al., 1998*) and are encoded by the CACNA1G ($Ca_V3.1$), CACNA1H ($Ca_V3.2$), and CACNA1I ($Ca_V3.3$) genes. The $Ca_V3$s have similar channel activation and inactivation kinetics and are ubiquitously expressed in the nervous, neuroendocrine, reproductive, and cardiovascular systems (*Perez-Reyes, 2003*; *Hansen, 2015*). In heterologous expression systems, $Ca_V3.3$-mediated currents display the slowest activation and inactivation kinetics and recover faster from inactivation (*Kozlov et al., 1999*), however, due to these relatively small differences, they are nearly indistinguishable from the other T-type isoform in native tissue. In the brain, abundant CACNA1I transcripts display remarkable regional distribution and appear to prevail in distal dendrites (*Perez-Reyes, 2003*; *Lee et al., 1999*), where $Ca_V3.3$ currents mediate the major sleep-spindle pacemaker in the thalamus (*Astori et al., 2011*). Transgenic mice lacking $Ca_V3.3$ channels display severe alterations to sleep-spindle generator rhythmogenic properties posing this T-type channel isoform as a critical target in the study of brain function and development (*Astori et al., 2011*). However, the expression of $Ca_V3.3$ channels overlaps with that of $Ca_V3.1$ and/or $Ca_V3.2$, which together with the lack of robust, isoform-selective pharmacology has hampered the elucidation of their specific contributions to cellular physiology.

T-type calcium channels share the characteristic modular topology of other voltage-gated ion channels (VGICs) (*Catterall et al., 2005*) that consists of a voltage sensor (VS) module formed by transmembrane segments S1 through S4 and a pore module (PM) composed of the transmembrane segments S5 and S6 connected by a re-entrant pore loop. The VS controls channel opening in response to changes in membrane potential and the PM provides aqueous passage for ions across the lipid membrane. The tetrameric arrangement of four PMs lining the permeation pathway surrounded by four VSs in either swapped or non-swapped configuration enables VGIC function (for review, see *Barros et al., 2019*). In voltage-gated potassium ($K_V$) and transient receptor potential (TRP) channels, each monomer (1 × VS + 1 × PM) is encoded by a core α-subunit; whereas in voltage-gated sodium ($Na_V$) and $Ca_V$ channels, the α-subunit contains the four homologous, but not identical, domains (DI-DIV) joined through large intracellular linkers (*Catterall, 2000*; *Catterall et al., 2005*).

Natural compounds that evolved to occlude ion channel's PM or to interact with their VS are distinguished broadly as pore blockers and gating modifiers, respectively. In VGICs, the extracellularly exposed areas of the VS are pharmacological targets of neurotoxins and synthetic compounds where at least three distinct pharmacological sites have been described in $Na_V$ channels (*Catterall et al., 2007*). Within the VS is a conserved S3b-S4 paddle motif which can be transplanted into the VS of other VGICs and retain toxin sensitivity (*Bosmans et al., 2008*). α-Scorpion toxins, sea-anemone toxins, and numerous spider toxins interact with $Na_V$ channel site 3, located in the extracellular loop between DIV-S3 and DIV-S4 thereby interfering with the conformational changes that couple channel activation to fast inactivation (*Hanck and Sheets, 2007*). Binding of β-scorpion toxins to the S1-S2 and S3-S4 loops of domain II (site 4) shifts the voltage dependence of channel activation towards depolarized voltages reducing the maximal current at normal activation potentials. Lastly, pharmacological site 6 (located near site 3) is targeted by the δ-conotoxins that slow $Na_V$ channel inactivation (*Terlau et al., 1996*). All these gating modifier peptides (GMPs) appear to 'hold' the VS in different conformations leading to their mechanism of modulation to be recognized as 'VS trapping' (*Cestèle et al., 1998*), a phenomenon also observed in the interaction of theraphotoxins with $K_V2.1$ (*Swartz and MacKinnon, 1997*) and agatoxins with $Ca_V$ channels (*McDonough et al., 1997a*).

Sequence and functional conservation between the VSs lead to promiscuous interactions between peptides and small molecules across VGIC families. Examples of these include ProTx-I (*Thrixopelma pruriens*, $Na_V/K_V/Ca_V$/TRPA1) (*Bladen et al., 2014*; *Bosmans et al., 2008*; *Gui et al., 2014*; *Middleton et al., 2002*); ProTx-II (*T. pruriens*, $Na_V/Ca_V$) (*Bladen et al., 2014*; *Middleton et al., 2002*); Kurtoxin (*Parabuthus transvaalicus*, $Na_V/Ca_V$) (*Chuang et al., 1998*); Hanatoxin (*Grammostola spatulata*, $K_V/Na_V/Ca_V$) (*Bosmans et al., 2008*; *Li-Smerin and Swartz, 1998*; *Swartz and MacKinnon, 1997*), and Pm1a (*Pelinobius muticus*, $Na_V/K_V$) (*Finol-Urdaneta et al., 2022*). Furthermore, small molecules such as capsaicin, capsazepine (TRPV1/$K_V/Ca_V$) (*Caterina et al., 1997*; *Kuenzi and Dale, 1996*; *McArthur*

*et al., 2019*), and A803467 (Na$_V$, Ca$_V$) (*Bladen and Zamponi, 2012*) amongst others are known to interact across VGIC families.

Shared ancestry and sequence conservation within the voltage sensing machinery have been used to rationalize commonalities in structure, gating kinetics, and pharmacophores between Na$_V$ and T-type channels (*Bladen and Zamponi, 2012*). Several Na$_V$-active GMPs were shown to inhibit Ca$_V$3.1 channels through interactions with the channel's domain III (Ca$_V$3.1$^{DIII}$) (*Salari et al., 2016*); whereas the potent Na$_V$1.7 inhibitor, μ-theraphotoxin Pn3a (*Pamphobeteus nigricolour*) (*Deuis et al., 2017*), also interacts with HVA Ca$_V$ channels (*McArthur et al., 2020*).

In this study, we have used whole-cell patch clamp electrophysiology, mutagenesis, and computational docking to probe Pn3a's interactions with the LVA Ca$_V$3 channels. Our results support the use of Pn3a as a molecular tool for the study of Ca$_V$3.3-mediated currents in native cells and highlight a previously unrecognized pharmacophore that may enable selective targeting of T-type channel isoforms.

## Results

### Pn3a selectively inhibits Ca$_V$3.3 channels

Functional assessment of Pn3a activity was examined on depolarization-activated calcium currents (I$_{Ca}$) through the human T-type calcium channel isoforms: Ca$_V$3.1, Ca$_V$3.2, and Ca$_V$3.3. Whole-cell currents were elicited by a 100 ms test pulse to −20 mV from a holding potential (Vh) of −90 mV at a frequency of 0.2 Hz and recorded at room temperature (20–22°C, *Figure 1A*). Pn3a (10 μM) strongly inhibited Ca$_V$3.3-mediated currents (90.2% ± 1.9%, n=5) with negligible effects over the two other T-type isoforms (Ca$_V$3.1: 3.8% ± 1.8%, n=5; Ca$_V$3.2: 2.5% ± 2.5%, n=5) (*Figure 1A and B*). These results indicate that Pn3a has >100-fold preference for Ca$_V$3.3 over the other highly homologous Ca$_V$3 isoforms. Scaled Ca$_V$3.3 macroscopic currents recorded in the absence (control) and presence of Pn3a (3 μM) display similar macroscopic activation kinetics ($\tau_{act, control}$ = 5.95 ± 0.54 ms vs. $\tau_{act, Pn3a}$ = 6.37 ± 0.44 ms; p=0.32, n=5, paired t-test), whereas Pn3a slowed macroscopic inactivation kinetics ($\tau_{inact control}$ = 39.91 ± 8.25 ms vs. $\tau_{inact, Pn3a}$ = 60.25 ± 16.02 ms; p=0.04, n=5, paired t-test) (representative currents provided in the inset, *Figure 1A*). The preferential actions of Pn3a highlight its potential as a molecular probe to isolate and study the contribution of Ca$_V$3.3 currents in native cells.

Pn3a inhibitory potency against Ca$_V$3.3 currents was assessed at increasing peptide concentrations from which a concentration-response curve was built (*Figure 1C*). Fit to a standard Hill equation rendered an IC$_{50}$ value of 0.96±0.05 μM (nH 0.87±0.04, n=5 per concentration) for the inhibition of Ca$_V$3.3 channels. This Hill coefficient is consistent with a 1:1 stoichiometry between Pn3a toxin and Ca$_V$3.3 channels.

The change in Ca$_V$3.3 peak current amplitude during Pn3a *washin* and *washout* enables the assessment of Pn3a binding to Ca$_V$3.3 channels. The inhibition of Ca$_V$3.3 I$_{Ca}$ by Pn3a was mono-exponential with progressively faster time constants ($\tau_{obs}$) at increasing peptide concentration (*Figure 1D*). The on- and off-rate constants (k$_{on}$ and k$_{off}$) were determined from the fit to the linear plot of 1/$\tau_{obs}$ vs. [Pn3a] (*Figure 1D*, n=5 per concentration), where k$_{on}$ is the slope and k$_{off}$ is the y-intercept. The linear regression line results in a k$_{on}$ of 0.036 μM$^{-1}$ s$^{-1}$ and k$_{off}$ of 0.052 s$^{-1}$ which yield a K$_D$ of 1.4 μM, in close agreement with the IC$_{50}$ value obtained (*Figure 1C*). k$_{off}$ was confirmed by fitting the *washout* to a single exponential (k$_{off}$ = 0.055 ± 0.002 s$^{-1}$). Hence, Pn3a inhibits Ca$_V$3.3-mediated currents without apparent actions on Ca$_V$3.1 or Ca$_V$3.2 exposed to up to 10 μM peptide.

### Pn3a modifies the gating of Ca$_V$3.3

The voltage dependence of Ca$_V$3.3 activation, deactivation, inactivation, and recovery from inactivation were investigated in the absence and presence of Pn3a (3 μM) (*Figure 2* and *Table 1*). Ca$_V$3.3 activation is shifted ~13 mV to more depolarized potentials (control V$_{0.5}$ = −28.3±0.4 mV, n=5, vs. Pn3a V$_{0.5}$ = −15.3±0.4 mV, n=5; p<0.0001), indicating that a stronger depolarization is required to enable channel opening in the presence of Pn3a (*Figure 2A and C*). The voltage dependence of Ca$_V$3.3 steady-state inactivation (SSI) was not affected by exposure to the spider peptide (control V$_{0.5}$ = −56.3±0.3 mV, n=5, vs. Pn3a V$_{0.5}$ = −56.6±0.4 mV, n=5, p=0.57, *Figure 2B–C*). Consistent with the observed slowing of inactivation, Ca$_V$3.3-mediated currents inactivated by a 200 ms pre-pulse to –20 mV recovered faster in the presence of Pn3a ($\tau$ =0.22 ± 0.01 s, n=5) than under control conditions

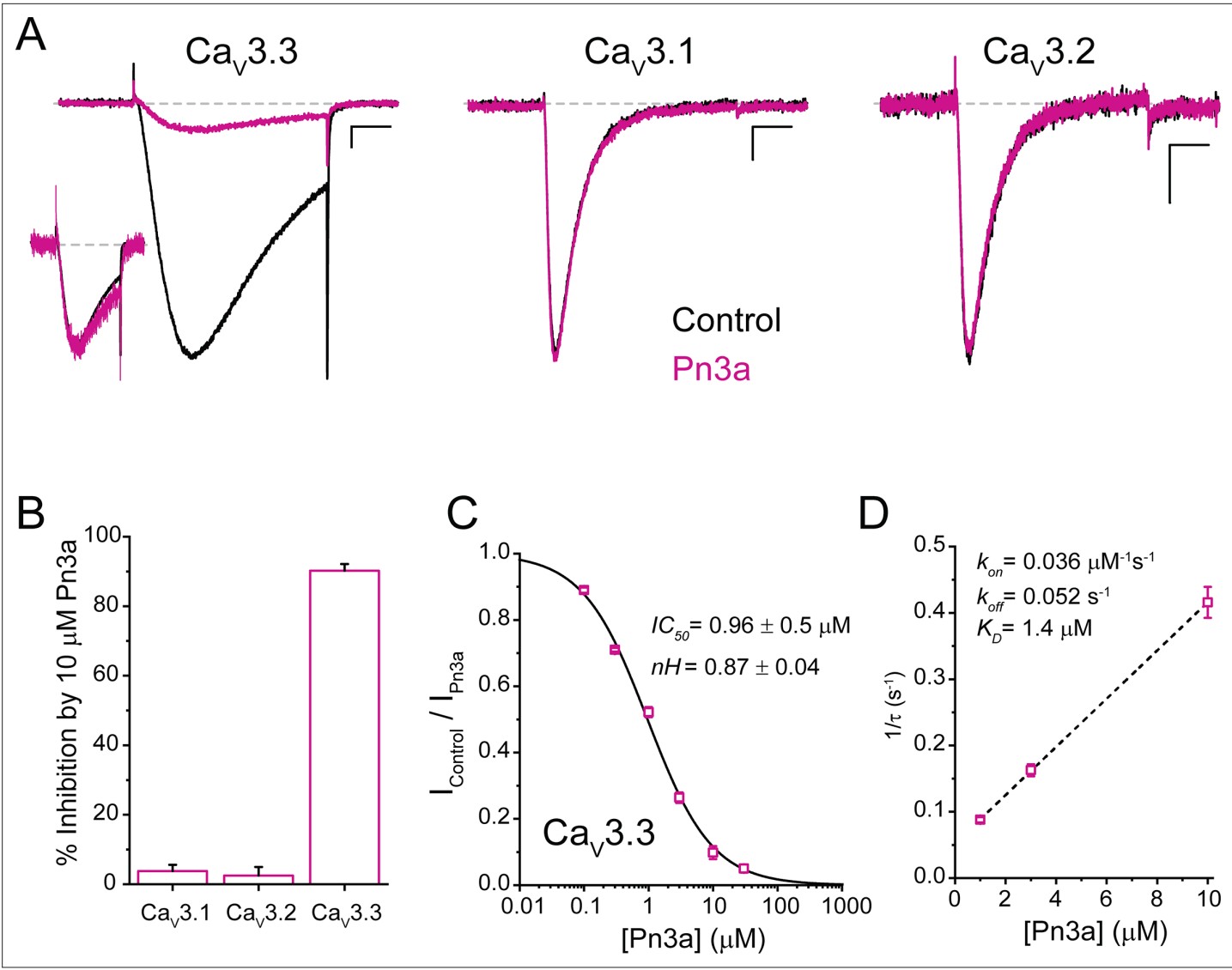

**Figure 1.** Pn3a preferentially inhibits human $Ca_V3.3$-mediated $Ca^{2+}$ currents. (**A**) $Ca_V3.3$ (left), $Ca_V3.1$ (middle), and $Ca_V3.2$ (right) currents elicited by 100 ms step depolarization to −20 mV (Vh −90 mV, 0.2 Hz) in the absence (control, black) and presence of 3 µM Pn3a (pink). Scale bars: 0.2 nA, 20 ms. The inset shows scaled $Ca_V3.3$ currents in control and in the presence of Pn3a with similar kinetics. (**B**) Bar graph summarizing % inhibition by 10 µM Pn3a of the three $Ca_V3$ isoforms. (**C**) Concentration-response curve for Pn3a inhibition of $Ca_V3.3$ currents. (**D**) Kinetics of Pn3a inhibition of $Ca_V3.3$. $K_{obs}$ was determined at three concentrations and fit with a linear equation where $K_{obs} = k_{on} \cdot [Pn3a] + k_{off}$. Data shown as mean ± SEM (n=5).

The online version of this article includes the following source data for figure 1:

**Source data 1.** Pn3a inhibition of T-type calcium channels.

($\tau$ = 0.34 ± 0.01 s, n=5; p<0.0001) (*Figure 2D–E*), suggesting a potential Pn3a-dependent destabilization of the inactivated state.

The voltage dependence of Pn3a inhibition of $I_{Ca}$ and $Na^{+}$-mediated ($I_{Na}$) $Ca_V3.3$ currents was examined at various test potentials (*Figure 3A* and *Figure 3—figure supplement 1*). Pn3A displayed a decrease in $Ca_V3.3$ $I_{Ca}$ inhibition at more depolarized potentials consistent with Pn3a preferentially binding to the down-state of the VS. To examine the voltage dependence across a larger voltage range, extracellular $Ca^{2+}$ was removed, permitting $Na^{+}$ to be the primary charge carrier through the open $Ca_V3.3$. Similar to that seen for $I_{Ca}$, $I_{Na}$ showed a similar voltage dependence, with stronger inhibition at more negative potentials (*Figure 3—figure supplement 1*).

The time course of tail current decay reflects the rate of channels leaving the open state (test potential) and entering the closed state (deactivation) upon return to the holding potential. We

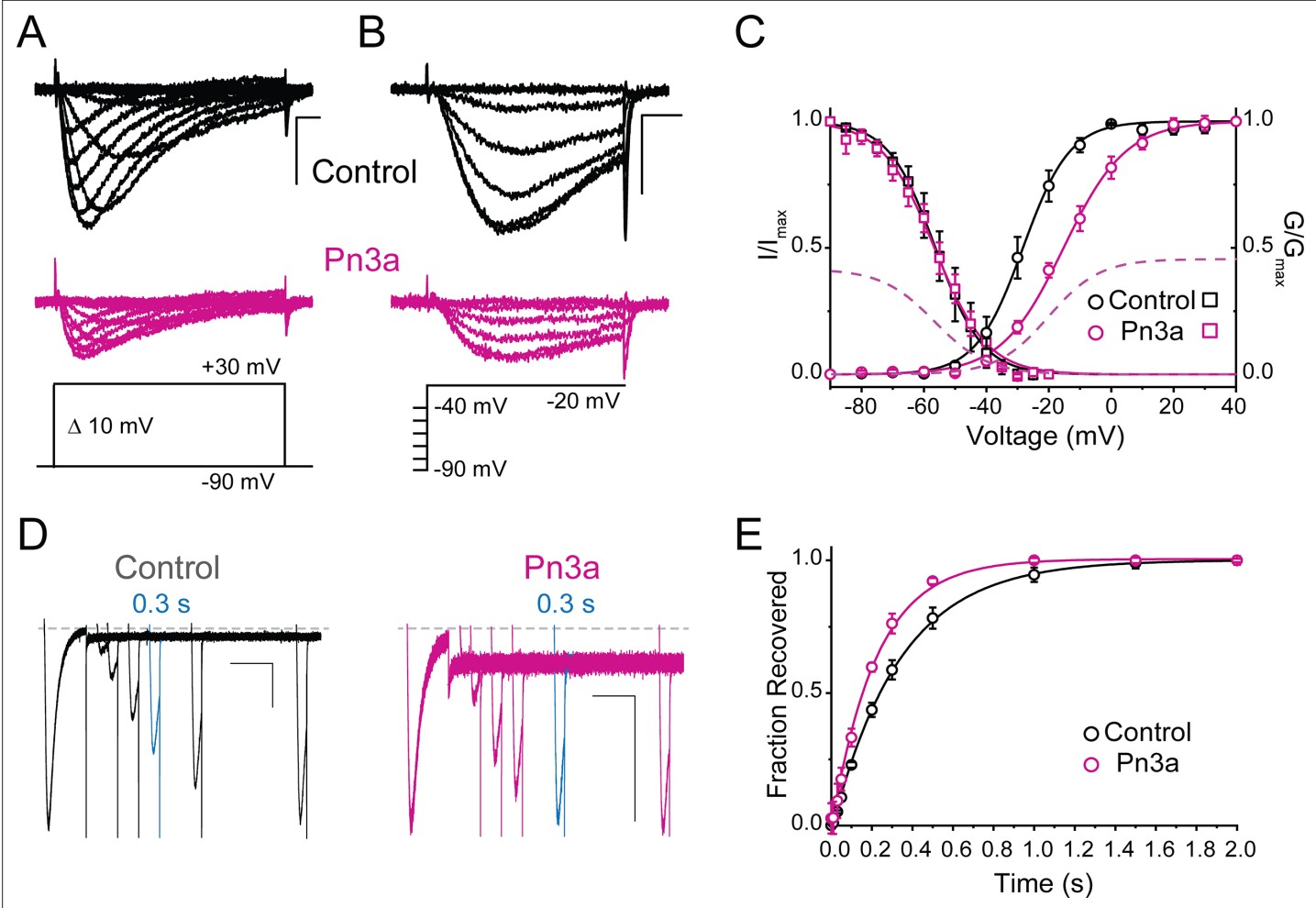

**Figure 2.** Pn3a produces a depolarizing shift in the voltage dependence of activation and speeds up recovery from inactivation of $Ca_V3.3$. (A–C) Effect of Pn3a on the voltage dependence of $Ca_V3.3$. Representative currents from (**A**) activation or (**B**) steady-state inactivation protocols, in the absence (top: control, black) and presence of 3 μM Pn3a (middle: Pn3a, pink) using the standard protocols (bottom). Scale bars: 0.5 nA, 10 ms. (**C**) Activation (circles) and steady-state inactivation (squares) relationships for $Ca_V3.3$ in the absence (black) and presence of 3 μM Pn3a (pink). Dashed lines represent non-normalized activation and inactivation curves of Pn3a. (**D**) Representative recovery from inactivation currents in control (left) and presence of 3 μM Pn3a (right) (trace shown in blue highlights the current recovered after 0.3 s). Scale bars: 0.2 nA, 200 ms. (**E**) Recovery from inactivation in the absence (black) and presence of 3 μM Pn3a (pink). Data shown as mean ± SEM (n=5).

The online version of this article includes the following source data and figure supplement(s) for figure 2:

**Source data 1.** Pn3a voltage effects o activation, steady-state inactivation, and recovery from inactivation.

**Figure supplement 1.** Current-voltage relation of control (black) and 3 μM Pn3a (pink) inhibition of human $Ca_V3.3$.

examined the modulation of $Ca_V3.3$ channel deactivation through tail current kinetic analysis. Pn3a-modified $Ca_V3.3$ currents displayed a faster deactivation time constant ($\tau_{deactivation}$) across all potentials tested compared to control (*Figure 3B–C*). This data suggests that Pn3a binding may destabilize the open state or stabilize the closed one. This results that Pn3a is a $Ca_V3.3$ GMP that decreases channel availability by increasing the energy required for channel opening.

## Pn3a interacts with $Ca_V3.3^{DII}$ S3-S4 paddles

To ascertain Pn3a's pharmacophore on $Ca_V3.3$ channels, we applied the chimeric approach of transplanting its four S3-S4 paddles into a $K_V$ channel based on a similar template to that previously described for $K_V2.1/Ca_V3.1$ chimeras (*Salari et al., 2016*). We substituted portions of the S3b-S4 from $Ca_V3.3$ DI to DIV into the $K_V1.7$ channel backbone. The sequence alignment of $Ca_V3.3$ DI-DIV S3-S4 segments and the corresponding extracellular region of $K_V1.7$ is presented in *Figure 4A*.

**Table 1.** Activation and inactivation values of $Ca_V3.3$ and $K_V1.7$-$Ca_V3.3^{DI-IV}$ chimers in the absence (control) and presence of Pn3a (3 µM for $Ca_V3.3$; 1 µM for $K_V1.7$-$Ca_V3.3^{D1-IV}$ chimeras).

| | | Control | Pn3a |
|---|---|---|---|
| hCa_V3.3 | $V_{0.5}$ activation | −28.3±0.4 mV (5) | **−15.3±0.4 mV (5) \*** |
| | ka Slope factor | 7.5±0.3 (5) | **10.0±0.3 (5) \*** |
| | $V_{0.5}$ SSI | −56.3±0.3 mV (5) | −56.6±0.4 mV (5) |
| | ka Slope factor | 7.5±0.2 (5) | 8.4±0.4 (5) |
| | τ Recovery | 0.34±0.01ms (5) | **0.22±0.01 (5) \*** |
| K_V1.7 | $V_{0.5}$ activation | −10.5±0.6 mV (7) | - |
| | ka Slope factor | 9.9±0.5 (7) | - |
| K_V1.7-Ca_V3.3^{DI} | $V_{0.5}$ activation | 91.3±0.4 mV (5) | 88.2±0.3 mV (5) |
| | ka Slope factor | 1.0±0.3 (5) | 11.8±0.2 (5) |
| K_V1.7Ca_V3.3^{DII} | $V_{0.5}$ activation | 97.3±0.4 mV (9) | **112.6±0.4 mV (7) \*** |
| | ka Slope factor | 14.1±0.3 (9) | **12.3±0.4 (7) \*** |
| K_V1.7-Ca_V3.3^{DIII} | $V_{0.5}$ activation | 71.3±0.8 mV (7) | 74.7±0.9 mV (5) |
| | ka Slope factor | 21.1±0.7 (7) | 22.0±0.8 (5) |
| K_V1.7-Ca_V3.3^{DIV} | $V_{0.5}$ activation | −60.8±0.8 mV (5) | −60.8±0.4 mV (5) |
| | ka Slope factor | 10.1±0.7 (5) | 10.0±0.4 (5) |

\*Significant determined from paired t-test with significance threshold set to p<0.05.

All four $K_V1.7/Ca_V3.3$ chimeric constructs were functional, mediating large $K^+$ currents that displayed distinct gating properties from those of the parental wild-type $K_V1.7$ ($V_{0.5}$ = −10.5±0.6 mV, n=7; **Figure 4B**, inset). Briefly, chimeric $K_V1.7/Ca_V3.3^{DI}$, $K_V1.7/Ca_V3.3^{DII}$ and $K_V1.7/Ca_V3.3^{DIII}$ displayed 80–100 mV depolarizing shifts in channel activation ($K_V1.7/Ca_V3.3^{DI}$ $V_{0.5}$=91.3 ± 0.4 mV, n=5; $K_V1.7/Ca_V3.3^{DII}$ $V_{0.5}$=97.3 ± 0.4 mV, n=9; $K_V1.7/Ca_V3.3^{DIII}$ $V_{0.5}$=71.3 ± 0.8 mV, n=7, two-way ANOVA, p<0.05), whereas $K_V1.7/Ca_V3.3^{DIV}$ activation was ~50 mV hyperpolarized ($K_V1.7/Ca_V3.3^{DIV}$ $V_{0.5}$ = −60.8±0.8 mV, n=5; two-way ANOVA, p<0.05) (**Figure 4**). Our results are in broad agreement with previous reports

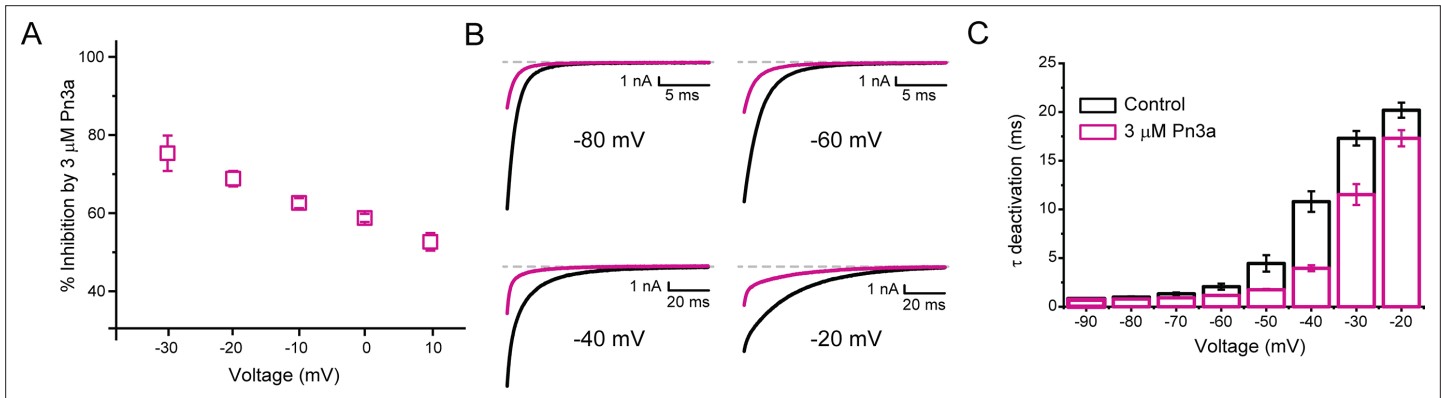

**Figure 3.** Pn3a inhibition of $Ca_V3.3$ is stronger at hyperpolarized potentials and speeds up channel deactivation. (**A**) Voltage dependence of Pn3a inhibition of $Ca_V3.3$-mediated $Ca^{2+}$ currents. (**B**) Representative $Ca_V3.3$ tail currents at different voltages in the absence (black) and presence (pink) of 3 µM Pn3a. (**C**) Summary of the time constant ( τ ) of $Ca_V3.3$ deactivation upon return to the holding potential (−90 mV) plotted against the activating (pre-pulse) potential in the absence (control, black) and the presence of Pn3a (3 µM, pink). Data shown as mean ± SEM (n=5).

The online version of this article includes the following source data and figure supplement(s) for figure 3:

**Figure supplement 1.** Voltage dependence of Pn3a inhibition of human $Ca_V3.3$ when $Na^+$ is the charge carrier.

**Source data 1.** Voltage dependence of Pn3a inhibition.

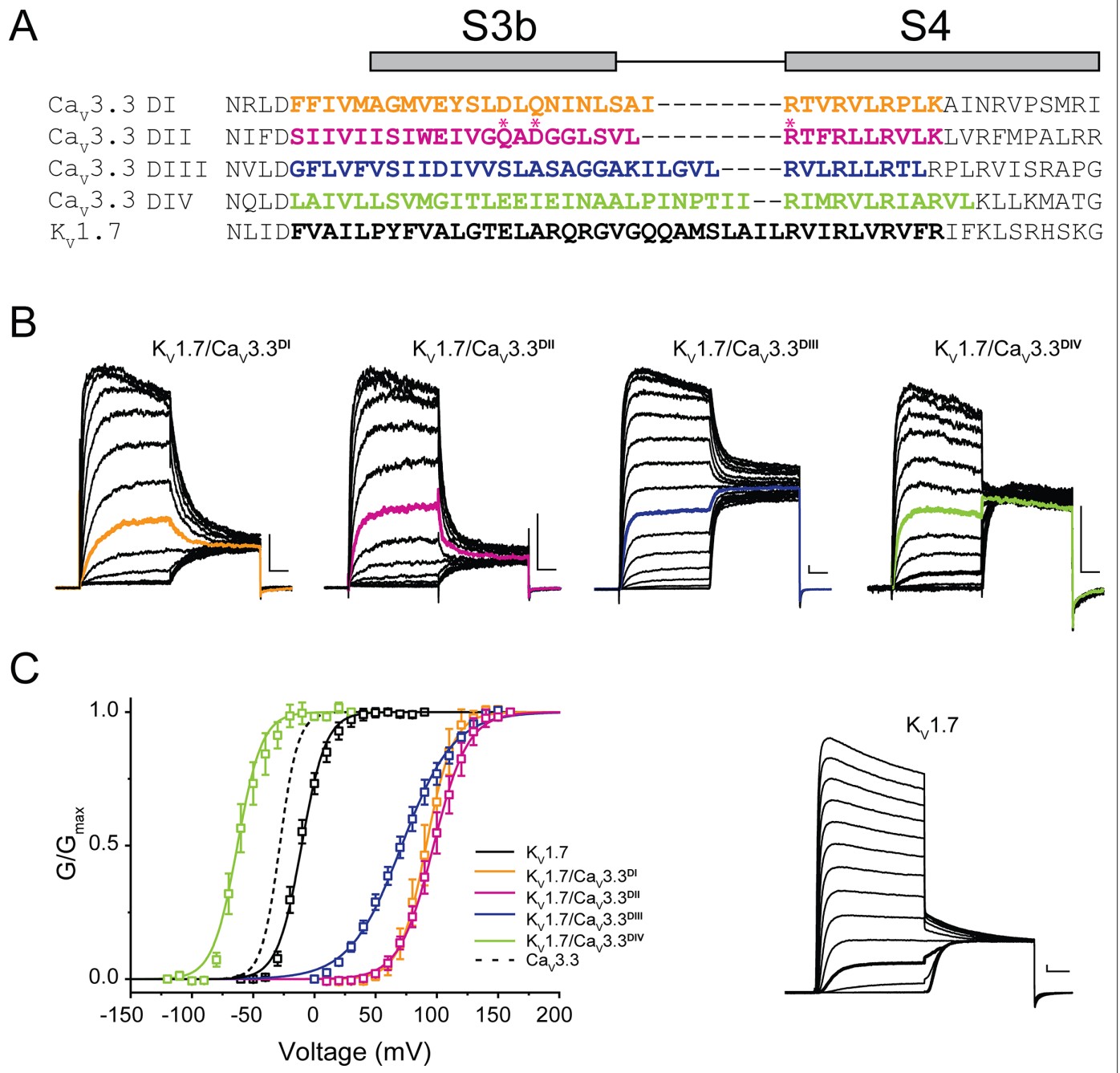

**Figure 4.** Chimeric constructs of the $K_V1.7$ channel with the $Ca_V3.3$ voltage sensor paddles. (**A**) Sequence alignment between paddle regions of $Ca_V3.3$ and $K_V1.7$. The coloured bolded sequences from $Ca_V3.3$ DI (yellow), $Ca_V3.3$ DII (pink), $Ca_V3.3$ DIII (blue), and $Ca_V3.3$ DIV (green) were grafted onto $K_V1.7$ (black). (**B**) Representative current traces in response to 50 ms long I-V protocols used to evaluate the voltage dependence of activation of all constructs (Vh = −80 mV). $K_V1.7/Ca_V3.3^{DI}$ (0–160 mV), $K_V1.7/Ca_V3.3^{DII}$ (0–160 mV), $K_V1.7/Ca_V3.3^{DIII}$ (0–160 mV) and $K_V1.7/Ca_V3.3^{DIV}$ (−120 to 30 mV). The traces highlighted in colour correspond to currents near half-activation potential ($V_{0.5}$). (**C**) Activation curves for all chimeras: $K_V1.7/Ca_V3.3^{DI}$ (yellow), $K_V1.7/Ca_V3.3^{DII}$ (pink), $K_V1.7/Ca_V3.3^{DIII}$ (blue), and $K_V1.7/Ca_V3.3^{DIV}$ (green) and the parental channel $K_V1.7$ (black). The dotted line corresponds to $Ca_V3.3$ activation for reference. The inset contains representative $K_V1.7$ currents (−60 to 90 mV). All scale bars: 1 nA, 10 ms.

for $Ca_V3.1$, where DIV displays a hyperpolarizing shift and DI-III show a variable depolarizing shift in $V_{0.5}$ (*Salari et al., 2016*).

$K_V1.7/Ca_V3.3^{DI-IV}$ chimeras and the parental $K_V1.7$ construct were analysed in control and after exposure to Pn3a (1 µM) and representative currents (Vh = −100 mV, with test pulse to +40 mV ($K_V1.7$), +120 mV ($K_V1.7/Ca_V3.3^{DI-III}$), or 0 mV ($K_V1.7/Ca_V3.3^{DIV}$)) in both conditions are included as

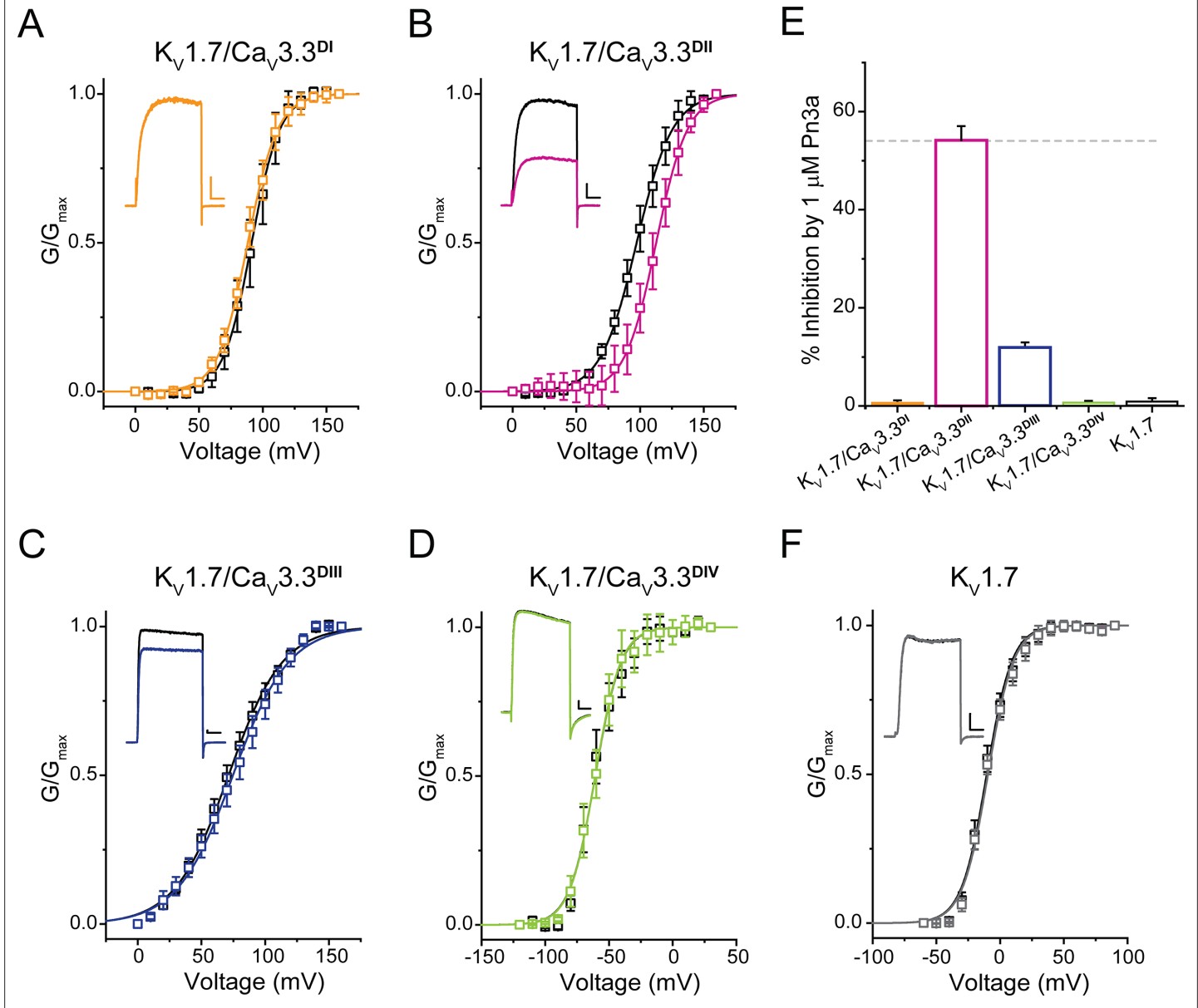

**Figure 5.** Pn3a shifts the activation of $K_V1.7/Ca_V3.3^{DII}$ chimeric channels. $G/G_{max}$-V relationships and representative currents obtained in the absence (control, black) and presence of 1 μM Pn3a (pink) for (**A**) $K_V1.7/Ca_V3.3^{DI}$ (90 mV), (**B**) $K_V1.7/Ca_V3.3^{DII}$ (100 mV), (**C**) $K_V1.7/Ca_V3.3^{DIII}$ (70 mV), and (**D**) $K_V1.7/Ca_V3.3^{DIV}$ (−60 mV). (**E**) Bar graph showing percent inhibition by 1 μM Pn3a of $K_V1.7/Ca_V3.3^{DI-IV}$ chimeras and $K_V1.7$ wt. (**F**) $K_V1.7$ $G/G_{max}$-V relationship and current traces obtained in the absence (control) and presence of Pn3a (−10 mV). All scale bars: 1 nA, 10 ms.

The online version of this article includes the following source data and figure supplement(s) for figure 5:

**Source data 1.** Voltage dependence of Pn3a inhibition of Kv1.7/Cav3.3 Chimera.

**Figure supplement 1.** Bar graph showing percent inhibition by 1 μM Pn3a of $K_V1.7/Ca_V3.3^{DII}$ chimera and mutants $K_V1.7/Ca_V3.3^{DII}$-Q260A, $K_V1.7/Ca_V3.3^{DII}$-D262A, and $K_V1.7/Ca_V3.3^{DII}$-R269A (corresponding to $Ca_V3.3$ −Q687A, −D689A, and −R696A, respectively).

insets (*Figure 5*). Similar to the parental channel, peak currents of $K_V1.7/Ca_V3.3^{DI}$ and $K_V1.7/Ca_V3.3^{DIV}$ chimaeras were insensitive to Pn3a ($K_V1.7$: 1.0% ± 0.8%, n=6; $Ca_V3.3^{DI}$: 0.6% ± 0.5%, n=5, and $Ca_V3.3^{DIV}$: 0.4% ± 0.2%, n=5) (*Figure 5A, D and E*), with $K_V1.7/Ca_V3.3^{DIII}$ displaying modest peptide-dependent inhibition (12.0% ± 0.6%, n=6) (*Figure 5C*).

In contrast, currents mediated by $K_V1.7/Ca_V3.3^{DII}$ were significantly reduced (54.2% ± 2.9%, n=7) in the presence of Pn3a (*Figure 5B*). Furthermore, conductance-voltage relationships for each S3-S4 paddle chimera in the absence and presence of Pn3a were built from peak currents in response to

standard I-V protocols ($K_V1.7$ –60 to +90 mV; $K_V1.7/Ca_V3.3^{DI-III}$ 0 to +160 mV; and $K_V1.7/Ca_V3.3^{DIV}$ –120 to +30 mV; Vh = −100 mV, 0.2 Hz). The latter analysis revealed an ~15 mV rightward shift in the half voltage of activation ($V_{0.5}$) exclusively in the $Ca_V3.3^{DII}$ chimera when exposed to Pn3a (control $V_{0.5}$=97.3 ± 0.4 mV, n=9; vs. Pn3a $V_{0.5}$=112.6 ± 0.4 mV, n=7, p<0.0001), strongly suggesting that Pn3a interacts with the S3-S4 region of domain II in $Ca_V3.3$. These results suggest that the positive shift in activation $V_{0.5}$ observed in the full length and DII chimera likely underpins Pn3a's mechanism of $Ca_V3.3$ inhibition, highlighting the differences in pharmacological properties of each voltage-sensing module.

## Molecular determinants of Pn3a inhibition of $Ca_V3.3$

The critical role of DII was further supported by examining the effects of Pn3a on full-length $Ca_V3$ channel constructs where the whole DII was swapped between isoforms. Replacement of $Ca_V3.3^{DII}$ with either $Ca_V3.1^{DII}$ or $Ca_V3.2^{DII}$ resulted in partial ($Ca_V3.3/Ca_V3.1^{DII}$: 33% ± 4.1%, n=5, one-way ANOVA p<0.0001) or complete ($Ca_V3.3/Ca_V3.2^{DII}$: 2.2% ± 1.5%, n=5, one-way ANOVA p<0.0001) loss of inhibition compared to the full-length $Ca_V3.3$ (90.2 ± 1.9% n=5) in the presence of 10 µM Pn3a (*Figure 6E*). The reverse constructs where $Ca_V3.3^{DII}$ was grafted onto $Ca_V3.1$ ($Ca_V3.1/Ca_V3.3^{DII}$) or $Ca_V3.2$ ($Ca_V3.2/Ca_V3.3^{DII}$) afforded Pn3a sensitivity to the parental $Ca_V3$ channel such that currents mediated by $Ca_V3.1/Ca_V3.3^{DII}$ were inhibited by 71.4% ± 4.1% (n=5), and $Ca_V3.2/Ca_V3.3^{DII}$ by 68.0% ± 3.6% (n=5) (*Figure 6F*). Thus, verifying that $Ca_V3.3^{DII}$ is required for Pn3a interaction with T-type calcium channels.

Molecular docking was used to assess the most energetically favoured binding poses for Pn3a on the extracellular-facing pockets of DII of the three channel isoforms, $Ca_V3.3$, $Ca_V3.2$, and $Ca_V3.1$ (*Figure 6* and *Figure 6—figure supplements 1–3*). Comparative analyses of toxin interactions with the other T-type calcium channel members showed that Pn3a forms fewer contacts with the DII S3-S4 linker of $Ca_V3.1$ and $Ca_V3.2$, compared to $Ca_V3.3$ (*Figure 6D*). Evaluation of the inter-residue contacts between the docked Pn3a and $Ca_V3.3^{DII}$ revealed several receptor residues predicted to form close contacts with the toxin (*Figure 6D*, upper left inset). In particular, Pn3a-K17 is predicted to be in close proximity to D689 and may form an especially strong interaction due to the possibility of a salt bridge contact. Guided by the docking results, point mutations of selected $Cav3.3^{DII}$ amino acids (E628A, Q687A, D689A, R696A, K1294A, N1310A, and Q1362A) were explored within full-length $Cav3.3$ channels. In comparison to wild-type $Ca_V3.3$, $Ca_V3.3$-D689A (27.6% ± 1.8%, n=5, one-way ANOVA p<0.0001) and $Ca_V3.3$-E628A (68.2% ± 2.0%, n=5, one-way ANOVA p<0.0001) displayed significant reductions in Pn3a inhibition (*Figure 6F*), whereas the peptide activity in the other mutants studied ($Ca_V3.3$-Q687A: 83.6% ± 0.5%, n=5; $Ca_V3.3$-R696A: 83.6% ± 2.7%, n=5; $Ca_V3.3$-K1294A: 84.8% ± 1.2%, n=5; $Ca_V3.3$-N1310A 79.2% ± 3.5%, n=5; and $Ca_V3.3$-Q1362A 85.4% ± 1.3%, n=5) was unaffected. The relevance of $Ca_V3.3$-D689 interaction with Pn3a was further substantiated by the loss of inhibition in the analogous charge neutralization $K_V1.7/Cav3.3^{DII}$ chimeric construct $K_V1.7/Cav3.3^{DII}$-D262A (*Figure 5—figure supplement 1*), thus pinpointing key molecular determinants responsible for Pn3a's selective inhibition of $Ca_V3.3$.

# Discussion

The present study demonstrates the functional activity of Pn3a as a gating modifier inhibitor of human $Ca_V3.3$ channels with >100-fold higher activity over $Ca_V3.1$ and $Ca_V3.2$. Our chimeric approaches revealed Pn3a's preference for $Ca_V3.3^{DII}$ S3-S4 paddle region, whereas comparative molecular docking amongst isoforms identified a novel binding site putatively determining Pn3a's proclivity towards $Ca_V3.3$. Thus, this investigation highlights Pn3a as the first molecular probe available for the study of $Ca_V3.3$ contribution in native cells. We propose that the unique features of Pn3a's $Ca_V3.3$ specificity may be exploited to design isoform-selective T-type calcium channel modulators.

## $Ca_V3$ subtype selectivity

µ-Theraphotoxin Pn3a was originally described as a potent $Na_V1.7$ inhibitor with an $IC_{50}$ of 0.9 nM, while also inhibiting other tetrodotoxin-sensitive $Na_V$ channels ($Na_V1.1$–1.4 and $Na_V1.6$) with 10- to 100-fold less potency (*Deuis et al., 2017*). More recently Pn3a was shown to inhibit murine dorsal root ganglion HVA $I_{Ca}$ and human $Ca_V1.2$, $Ca_V1.3$, $Ca_V2.1$, and $Ca_V2.2$ calcium channels expressed in HEK293T cells ($IC_{50}$ 3–10 µM) (*McArthur et al., 2020*). Perhaps a more interesting aspect of Pn3a

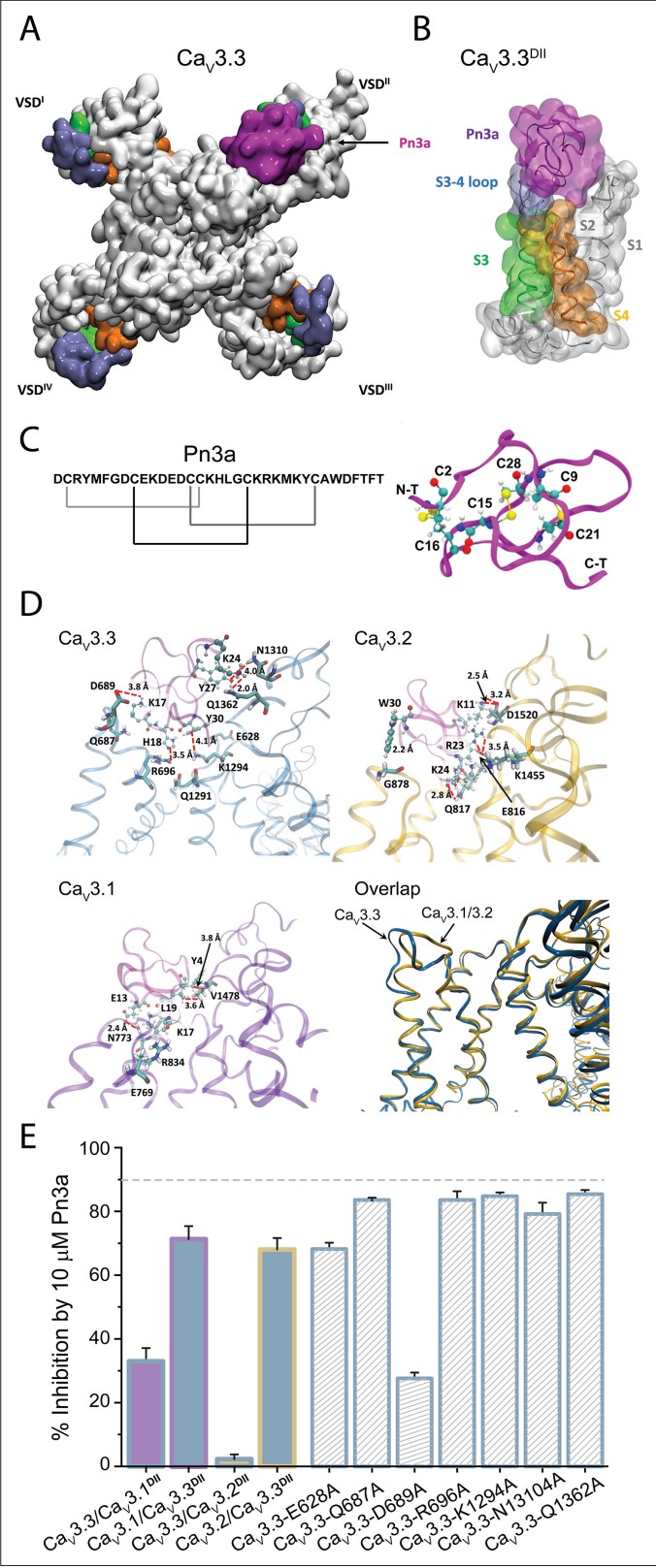

**Figure 6.** Pn3a and Ca$_V$3.3 interactions and subtype specificity. (**A**) Top view of Pn3a (magenta) binding to Ca$_V$3.3$^{DII}$. The S3 (lime), S3-4 loop (ice blue), and S4 (pink) are highlighted. (**B**) Side view of the binding conformation of Pn3a to Ca$_V$3.3$^{DII}$. (**C**) Left: Pn3a amino acid sequence and disulfide connectivity. Right: 3D structure of Pn3a (PDB ID: 5T4R *Deuis et al., 2017*) showing cysteine residues in CPK linked by salt bridges (yellow). (**D**) Pairwise amino acid

*Figure 6 continued on next page*

*Figure 6 continued*

interactions between Pn3 (magenta) with the extracellular loop of $Ca_V3.3^{DII}$ (top left, blue), $Ca_V3.2^{DII}$ (top right, yellow), and $Ca_V3.1^{DII}$ (bottom left, purple). Overlap of the Pn3a-bound ribbon structures of all $Ca_V3$ highlighting differences in extracellular S3-S4 loops between $Ca_V3.3$ and the other two $Ca_V3$ isoforms (bottom right). S1, S1-2 loop, and S2 were removed for clarity. (**E**) Bar graph showing percent inhibition by 10 µM Pn3a of $Ca_V3^{DII}$ domain-swapped $Ca_V3$ constructs and point mutants.

The online version of this article includes the following source data and figure supplement(s) for figure 6:

**Source data 1.** Pn3a docking coordinates for T-type calcium channels.

**Figure supplement 1.** The binding conformation of Pn3a at voltage-sensing domain (VSD)-II of T-type calcium channels.

**Figure supplement 2.** Minor interactions between the residues of Pn3a and the corresponding residues on $hCa_V3$ channels.

**Figure supplement 3.** Predicted binding affinity values for Pn3a binding to the DI, DII, DIII, and DIV domains of the voltage sensor (VS) modules on $hCa_V3.3$ calculated using Autodock Vina.

---

activity is its >100-fold selectivity for human $Ca_V3.3$ over both $Ca_V3.1$ and $Ca_V3.2$ (*Figure 1*) which distinguishes this spider peptide as a unique isoform-selective modulator of T-type calcium channels. Only a handful of venom-derived peptides have been shown to interact with T-type calcium channels. The scorpion peptide Kurtoxin was the first GMP reported to modulate $Ca_V3$ channels with a high affinity for $Ca_V3.1$ and $Ca_V3.2$ channels (*Chuang et al., 1998*). The closely related peptide KLI, from *Parabuthus granulatus*, was later shown to inhibit $Ca_V3.3$ with an $IC_{50} \sim$ 450 nM (*Olamendi-Portugal et al., 2002*). However, the T-type isoform selectivity and binding site of these two peptides were not comprehensively documented at the time. The activity of spider Protoxins I and II against the three $Ca_V3$ channels revealed that ProTx-I preferentially modulates $Ca_V3.1$ channels, whereas ProTx-II targets $Ca_V3.2$ (*Bladen et al., 2014*; *Salari et al., 2016*). Thus, Pn3a complements the molecular toolbox for the study of T-type calcium channels in native tissues.

## Voltage dependence and current kinetics

Pn3a inhibits $Ca_V3.3$ by inducing a depolarizing shift in $Ca_V3.3$ voltage dependence of activation in a manner analogous to Kurtoxin, Protoxin I, and Protoxin II actions on $Ca_V3.1$ channels (*Edgerton et al., 2010*; *Chuang et al., 1998*). This is also consistent with Pn3a's modulation of $Na_V1.7$ (*Deuis et al., 2017*) but not of $Ca_V2.2$ channels for which a hyperpolarizing shift in the voltage dependence of inactivation was associated with current inhibition (*McArthur et al., 2020*). In contrast to the twofold slower recovery from inactivation reported for Pn3a-bound $Na_V1.7$ channels, currents mediated by Pn3a-modified $Ca_V3.3$ channels recover ~1.5-fold faster from inactivation (*Figure 2D/E*). This and the observed slowing of inactivation (at high Po) are suggestive of a toxin-induced destabilization of the inactivated state. Nevertheless, given Pn3a's voltage dependence of Cav3.3 inhibition (*Figure 3A*), voltage-dependent unbinding of the toxin at more positive potentials could also contribute to an apparent faster recovery from inactivation as reported for $Ca_V$ active toxins such as AgaIVA (*McDonough et al., 1997b*) and SNX482 (*Bourinet et al., 2001*).

Pn3a inhibition of $Ca_V3.3$ was voltage-dependent with greater inhibition at more negative potentials (*Figure 3A*) similar to Kurtoxin-inhibited $Ca_V3.1$ channel currents (*Chuang et al., 1998*). However, a delay in $Ca_V3.1$ activation kinetics was apparent in the presence of both Kurtoxin and ProTx-II (*Edgerton et al., 2010*; *Chuang et al., 1998*) but not in Pn3a-modified $Ca_V3.3$. The spider peptides Pn3a (this study) and ProTx-II (*Edgerton et al., 2010*) appear to slow T-type channel deactivation suggestive of stabilization of the channel's closed state, whereas Kurtoxin, from scorpion venom, does not affect $Ca_V3.1$ channel closure (*Chuang et al., 1998*) highlighting incompletely understood aspects of GMP/VGIC interactions.

## Interaction of Pn3a with $Ca_V3.3$ VS paddles

The portability of the S3-S4 paddle region was shown more than 20 years ago (*Swartz and MacKinnon, 1997*). To date, most studies, if not all, have used the $K_V2.1$ channel backbone for the identification of GMPs binding sites in $Na_V$ and $Ca_V$ channels expressed in *Xenopus* oocytes (*Bosmans et al., 2008*; *Salari et al., 2016*). For our chimera studies, we selected the $K_V1.7$ backbone given

the remarkable scarcity of GMPs interacting with $K_V1$ channels (*Finol-Urdaneta et al., 2020*). The robust expression of this channel in heterologous systems (*Finol-Urdaneta et al., 2006*; *Finol-Urdaneta et al., 2012*) and resistance to Pn3a modulation (*Figure 5E/F*) make it suitable to assess peptide binding to $Ca_V3.3$ paddle motifs in mammalian cells. The generated $K_V1.7/Ca_V3.3$ constructs resulted in four functional, voltage-gated chimeric potassium channels that exhibited distinct gating properties to those of the parental scaffold reflecting the acquisition of the grafted S3-S4 paddle regions from DI to DIV of $Ca_V3.3$. Namely, $K_V1.7/Ca_V3.3^{DI-DIII}$ all displayed >80 mV depolarizing shifts in channel activation compared to wild-type $K_V1.7$, whereas the $Ca_V3.3^{DIV}$ bearing chimeric construct presented a comparable magnitude shift in the hyperpolarizing direction (*Figure 4*).

Analogous chimeric approaches of $K_V2.1/Ca_V3.1D^{I-IV}$ have indicated that ProTx-II, PaTx-1, GsAF-I, and GsAF-II exert their inhibitory actions predominantly through interaction with $Ca_V3.1^{DIII}$ (*Salari et al., 2016*), whereas Pn3a does so by targeting $K_V2.1/Na_V1.7^{DII}$ and $K_V2.1/Na_V1.7^{DIV}$ (*Deuis et al., 2017*). Here, we observe modest Pn3a inhibition of $K_V1.7/Ca_V3.3^{DIII}$-mediated currents and potent effects on $K_V1.7/Ca_V3.3^{DII}$ chimeras with current inhibition coupled to toxin-induced rightward shift in the voltage dependence of activation as evidence of preferential interactions with $Ca_V3.3^{DII}$ (*Figure 5B/F*). A substantial body of literature has shown that DI-DIII and DIV are important for $Na_V$ activation and inactivation, respectively (*Ahern et al., 2016*). Thus, the predominant effects of Pn3a on $Ca_V3.3$ activation are consistent with its interaction with DII of this channel.

Furthermore, it has been shown that ProTx-I inhibits $Ca_V3.3/Ca_V3.1^{DIV}$ chimeric channels while interacting less potently with $Ca_V3.3/Ca_V3.1^{DII}$. However, a clear binding site could not be delineated through mutation of individual $Ca_V3.1^{DII}$ residues as those did not result in measurable changes in toxin affinity (*Bladen et al., 2014*). Our domain swap experiments verified that Pn3a's actions are largely determined by $Ca_V3.3^{DII}$ from which aspartate in position 689 constitutes an important interaction site as shown by its alanine replacement in $Ca_V3.3$ and $K_V1.7/Ca_V3.3^{DII}$. It can be surmised that subtle, but significant GMP/VGIC interaction differences highlight incompletely understood idiosyncrasies related to molecular aspects of ion channel function between isoforms as well as how peptide interactions may be affected by the experimental manipulation and conditions used for their study.

## A novel binding site with $Ca_V3$ subtype selectivity

Our molecular docking calculations suggest that Pn3a binds within the groove formed between extracellular linkers S1-S2 and S3-S4 supported by electrostatic interactions with the $Ca_V3.3^{DII}$ S3-S4 paddle, which bears some similarity with the $Ca_V3.1/ProTx-II$ complex in which favourable binding sites occur in $Ca_V3.1$'s DII and DIV (*Bladen et al., 2014*).

The interaction of Pn3a with $Ca_V3.3$ channels involves substantial interactions within the S3-S4 paddle (K17-Q687) and S4 (H18-R696) of $Ca_V3.3^{DII}$, while a close contact may also exist between the sidechain of D689 and the sidechain of Pn3a-K17 (*Figure 6D*, upper left panel). The latter is consistent with the establishment of a salt bridge between these residues. The loss of Pn3a inhibition upon charge neutralization at this position lends support to this docking model.

The proximity between cationic residues (K17 and H18) on loop 3 of Pn3a and the $Ca_V3.3D^{II}$ S3-S4 paddle places this peptide at the hollow between S1-S2 and S3-S4 linkers in a binding conformation similar to other tarantula toxins targeting $Na_V$ channel site 4, like HwTX-IV, ProTx-II, and HNTX-III, that bear critical basic and hydrophobic amino acids within loop 4 that interact with acidic residues on the $Na_V^{DII}$'s S3-S4 linker (*Deng et al., 2013*; *Liu et al., 2013*; *Bosmans and Swartz, 2010*; *Figure 6—figure supplements 1 and 2*). Hence, the predicted Pn3a interaction with $Ca_V3.3$ appears to share overall similarities with previously identified spider peptide toxins modulating $Na_V$ and $K_V$ channels in which the common bioactive surface consists of positively charged and hydrophobic residues (*Smith et al., 2007*; *Corzo et al., 2005*).

The findings presented here establish Pn3a as a gating modifier modulator of $Ca_V3.3$ channels interacting with the paddle motif of DII's VS module through putative stabilization of the closed/resting state and concomitant channel current inhibition. Pn3a's >100-fold higher potency against $Ca_V3.3$ over the other two $Ca_V3$ isoforms is rationalized through recognition of a previously unknown drug binding site that may be exploited in the design of isoform-selective $Ca_V3$ channel modulators.

## Materials and methods

### Cell lines, culture, and transfections

Human embryonic kidney (HEK293T, authenticated by STR profiling and mycoplasm free) cells containing the SV40 Large T-antigen were cultured and transfected by calcium phosphate method as reported previously (*McArthur et al., 2018*). In brief, cells were cultured at 37°C, 5% $CO_2$ in Dulbecco's modified Eagle's medium (DMEM, Invitrogen Life Technologies, VIC, Australia), supplemented with 10% fetal bovine serum (FBS, Bovigen, VIC, Australia), 1% GlutaMAX and penicillin-streptomycin (Invitrogen). HEK293T cells were then transiently co-transfected with the different $Ca_V$ channel isoforms and green fluorescent protein (GFP) for visualization, using the calcium phosphate method. cDNAs encoding human $Ca_V3.1$ (provided by Dr G Zamponi), human $Ca_V3.2$ (a1Ha-pcDNA3 was a gift from Dr E Perez-Reyes, Addgene #45809) (*Cribbs et al., 1998*), human $Ca_V3.3$ (a1Ic-HE3-pcDNA3 also from Dr E Perez-Reyes, Addgene #45810) (*Gomora et al., 2002*) in combination with GFP. $Ca_V3^{DII}$ swap constructs and $K_V1.7/Ca_V3.3$ chimeras were custom synthesized by GeneScript, NJ.

Chinese hamster ovary (CHO) cells were used to express $K_V1.7$ and $K_V1.7-Ca_V3.3$ paddle chimeras. Cell culture conditions were the same as the HEK293T cells except DMEM was substituted with DMEM/F12 (Invitrogen). CHO cells were transfected with cDNAs encoding $K_V1.7$ and $K_V1.7/Ca_V3.3^{DI-DIV}$ chimeras using Lipofectamine 2000 (Invitrogen) as per the manufacturer's protocol and used for recordings 12–48 hr post-transfection.

### Electrophysiology

Whole-cell patch clamp configuration was used to record calcium ($I_{Ca}$) or potassium ($I_K$) currents in transiently transfected HEK293T cells. Recordings were made using a MultiClamp 700B amplifier, digitized with a DigiData1440, and controlled using Clampex11.1 software (Molecular Devices, San Jose, CA). Whole-cell currents were sampled at 100 kHz and then filtered to 10 kHz, with leak and capacitive currents subtracted using a −P/4 protocol for $Ca^{2+}$ currents and uncorrected for $K^+$ currents. All recordings were series compensated 60–80%. External solution for $I_{Ca}$ contained in mM: 100 NaCl, 10 $CaCl_2$, 1 $MgCl_2$, 5 CsCl, 30 TEA-Cl, 10 D-glucose and 10 HEPES, pH 7.3 with TEA-OH. External solution for $I_K$ contained in mM: 140 NaCl, 5 KCl, 1 $MgCl_2$, 2 $CaCl_2$, 10 glucose, and 10 HEPES, pH 7.3 with NaOH. Fire-polished borosilicate (1B150F-4, World Precision Instruments, Sarasota, FL) patch pipettes were used with resistance of 1–3 MΩ. Intracellular recording solution contained as follows (mM): 140 KGluconate, 5 NaCl, 2 $MgCl_2$, 5 EGTA, and 10 HEPES, pH 7.2 with KOH. Cells were continuously perfused with extracellular solution at a rate of 1.2 ml/min, while toxin application was superfused onto the cell through a capillary tube attached to a syringe pump (2 µl/min), directly onto the cell being recorded.

For experiments on $Ca_V3$s, all cells were held at −90 mV. To examine the onset of block, test pulses (100 ms, 0.5 Hz) to −20 mV were applied. To generate activation curves, cells were pulsed from −90 to +40 mV in 10 mV increments at 0.5 Hz. SSI curves were generated by measuring the peak current from a test pulse to −20 mV when preceded by a 1 s pre-pulse from −90 to 20 mV (0.1 Hz). Recovery from inactivation curves was produced by varying the time (0–2 s) between two depolarizing pules to −20 mV (P1 200 ms, P2 50 ms).

For experiments on $K_V1.7$ and $Ca_V3.3$ DI-IV chimeras, cells were held at −80 mV and activation curves were generated by applying a 50 ms pre-pulse to varying potentials depending on the channel construct examined ($K_V1.7$ −60 to 90 mV, DI-III 0 to +160 mV, and DIV −120 to +30 mV) followed by a 50 ms test pulse to measure tail currents ($K_V1.7$ 0 mV, DI-III +80 mV, and DIV −20 mV). To measure the onset of Pn3a inhibition, test pulses (50 ms) from a holding potential of −100 mV to a test potential determined for each channel construct ($K_V1.7$ 40 mV, DI-III 120 mV, and DIV 0 mV) were elicited at 0.2 Hz.

### Structures and homology modelling of human T-type calcium channels and the docking of Pn3a

The cryo-EM structures of human $Ca_V3.1$ (PDB ID: 6KZO) (*Zhao et al., 2019*) and $Ca_V3.3$ (PDB ID: 7WLI) (*He et al., 2022*) were used for docking calculations. The comparative model of human$Ca_V3.2$ was built upon the cryo-EM structure of h$Ca_V3.1$ in the apo form (PDB ID: 6KZO) via the SWISS-MODEL server (https://swissmodel.expasy.org/). The 3D structure of µ-theraphotoxin (TRTX)-Pn3a

(PDB ID: 5T4R) (*Deuis et al., 2017*) was docked into the *three* T-type calcium channels via Autodock Vina with a grid box of 40Å × 40Å × 40 Å. We also compared the binding mode and the binding affinity amongst the four VSs of $Ca_v3.3$, to identify which VS module Pn3a preferentially targets. The docking results were further analysed via Discovery Studio 2017R2 and visualized using Visual Molecular Dynamics (VMD) version 1.9.3 (*Humphrey et al., 1996*).

## Data and statistical analysis

All data analysis and graphs were generated in OriginPro (Origin Lab Corporation, Northampton, MA). Concentration-response curves were generated by plotting peak current amplitudes in the presence of Pn3a ($I_{Pn3a}$), over the current before Pn3a application ($I_{Control}$). The resulting curve was fit with a sigmoidal curve according to the following expression:

$$I_{Pn3a}/I_{Control} = 1 + [Pn3a]^n / \left( IC50^n + [Pn3a]^n \right) \tag{1}$$

where IC50 is the half-maximal inhibitory concentration and n is the Hill coefficient. Activation (*Equation 2*) and SSI (*Equation 3*) curves were fit by the modified Boltzmann equation:

$$I = 1 - 1/ \left( 1 + \exp \left( \frac{Vm - V_{0.5}}{ka} \right) \right) \tag{2}$$

$$G = 1/ \left( 1 + \exp \left( \frac{Vm - V_{0.5}}{ka} \right) \right) \tag{3}$$

where I is the current or G is the conductance, $V_m$ is the pre-pulse potential, $V_{0.5}$ is the half-maximal activation potential, and ka is the slope factor. Recovery from inactivation plots was fit using a single exponential of the following equation:

$$\left( \frac{P2}{P1} \right) = 1 + A_{fast} * \exp \left( -\frac{t}{t} \right) \tag{4}$$

where $\tau$ is the time constant and A is the amplitude. Statistical significance ($p < 0.05$) was determined using paired or unpaired t-test or two-way ANOVA followed by a Tukey multiple comparison test if F achieves the level of statistical significance of $p < 0.05$ and no variance inhomogeneity. All data are presented as mean ± SEM (n), where n is individual cells with all experimental results containing n≥5 individual cells.

## Acknowledgements

We are grateful to Alexander Mueller (PhD, Vetter laboratory at the Institute of Molecular Bioscience, University of Queensland) for synthesizing the μ-Theraphotoxin Pn3a used in this research. This work was supported by the Rebecca Cooper Foundation for Medical Research Project Grant (PG2019396) to JRM and the National Health and Medical Research Council (NHMRC) Program Grant (APP1072113) to DJA. JRM and RKF-U are grateful to Emma and Zack O Yepugas for continuous support. Computational resources were provided by the National Computational Infrastructure (NCI) which is funded by the Australian Government, and the Pawsey Supercomputing Centre which is funded by the Australian Government and the Government of Western Australia. We acknowledge PRACE for awarding access to the Piz Daint cluster at the Swiss National Supercomputing Centre (CSCS), Switzerland.

## Additional information

### Funding

| Funder | Grant reference number | Author |
| --- | --- | --- |
| Rebecca L. Cooper Medical Research Foundation | PG2019396 | Jeffrey R McArthur |

| Funder | Grant reference number | Author |
|---|---|---|
| National Health and Medical Research Council | APP1072113 | David J Adams |

The funders had no role in study design, data collection and interpretation, or the decision to submit the work for publication.

## Author contributions

Jeffrey R McArthur, Conceptualization, Resources, Data curation, Formal analysis, Funding acquisition, Validation, Investigation, Visualization, Methodology, Writing - original draft, Project administration, Writing - review and editing; Jierong Wen, Data curation, Formal analysis, Investigation, Visualization, Writing - review and editing; Andrew Hung, Conceptualization, Resources, Data curation, Software, Formal analysis, Supervision, Funding acquisition, Validation, Investigation, Visualization, Methodology, Writing - original draft, Project administration, Writing - review and editing; Rocio K Finol-Urdaneta, Conceptualization, Data curation, Visualization, Writing - original draft, Writing - review and editing; David J Adams, Conceptualization, Resources, Supervision, Funding acquisition, Project administration, Writing - review and editing

## Author ORCIDs

Jeffrey R McArthur ⓘ http://orcid.org/0000-0002-2546-7913
Andrew Hung ⓘ http://orcid.org/0000-0003-3569-2951
Rocio K Finol-Urdaneta ⓘ http://orcid.org/0000-0003-2157-4532
David J Adams ⓘ http://orcid.org/0000-0002-7030-2288

## Decision letter and Author response

Decision letter https://doi.org/10.7554/eLife.74040.sa1
Author response https://doi.org/10.7554/eLife.74040.sa2

# Additional files

## Supplementary files
• Transparent reporting form

## Data availability
All data generated or analysed during this study are included in the manuscript and supporting file.

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
