## [Editor Report]

Low voltage activated T-type calcium channels (CaV3.1-CaV3.3) are important to several physiological processes, but the functions of individual isoforms are difficult to distinguish pharmacologically. This study reports a spider toxin, Pn3a, which exhibits 100-fold selectivity for inhibiting CaV3.3 over CaV3.1 and CaV3.2 isoforms. This specificity is conferred by a specific interaction site on the Cav3.3 domain II voltage sensor.

---

## [Decision Letter]

**Decision letter after peer review:**

Thank you for submitting your article "µ-Theraphotoxin-Pn3a inhibition of Cav3.3 channels reveals a novel isoform-selective drug binding site" for consideration by *eLife*. Your article has been reviewed by 3 peer reviewers, one of whom is a member of our Board of Reviewing Editors, and the evaluation has been overseen by Richard Aldrich as the Senior Editor. The reviewers have opted to remain anonymous.

Essential revisions:

1) There are some complexities associated with interpretation of the data obtained using Kv1.7/CaV3.3 VSD chimeras. Chimeras between Cav3.1 and Cav3.3 where entire membrane domains were swapped are in the public domain. CaV3.1/CaV3.3 chimeras with VSDII swapped (or similar chimera) should be done to strengthen the idea that Pn3a inhibits CaV3.3 by binding to VSDII.

2) Several residues in CaV3.3 are proposed to mediate selective inhibition by Pn3a based on MD simulations. These predictions should be validated by mutagenesis and functional experiments.

3) Pn3a blocks various CaV1/2 channels and NaV1.7. The latter is particularly high affinity (nM range) compared to CaV3.3 (uM). This limitation is noted briefly in lines 314-316. The authors should expand this further for the general audience who may be interested in using this toxin practically.

*Reviewer #1 (Recommendations for the authors):*

1. It seems surprising that the scaled Pn3a-inhibited CaV3.3 currents displayed the same activation kinetics but slower inactivation compared to uninhibited channels (Figure 1). Intuitively, the inhibitory effect of Pn3a would appear to be more consistent with slower activation and faster inactivation kinetics. Single channel experiments would be needed to distinguish these two different interpretations which could both give rise to similar macroscopic currents.

2. The effects of Pn3a on CaV3.3 gating should be modeled to provide deepened insights into the possible mechanisms of action that can explain the data.

3. Does Pn3a inhibit the maximal CaV3.3 current amplitude? The use of normalized G-V curves obscures this important point. The impact of Pn3a on Gmax should be provided.

4. The computational docking model (Figure 6) should be validated by mutagenesis experiments.

5. A significant rationale for the paper was the idea that Pn3a could potentially be used to distinguish CaV3.3-mediated effects from those of co-expressed CaV3.3 and CaV3.2 in native tissues. This capability certainly would certainly be a significant achievement. This prediction would need to be tested in a native cell type expressing multiple CaV3 channel isoforms. However, previous reports that Pn3a also inhibits Nav1.7 and HVA Ca channels suggest that interpreting the effects of the toxin in native cells is likely to be highly complicated.

*Reviewer #2 (Recommendations for the authors):*

The electrophysiological analysis of CaV3.x comparing the effect of Pn3a is well done and convincing. As noted in the public review, the main weakness is that the mechanism by why Pn3a accomplishes this selectivity is not fully clear and the identification of specific residues that support Pn3a interaction relies solely on computational docking.

1. The mechanisms that confer subtype specificity of Pn3a for CaV3.3 is not clear. The docking simulation suggests that Pn3a interacts with both CaV3.2 and CaV3.3 at slightly distinct residues on the same, but only CaV3.3 is inhibited. Is there any experimental evidence that shows that Pn3a binds to CaV3.1/CaV3.2 VSDs? Also the docking seems to suggest that Pn3a interacts with all 4 VSDs of CaV3.3. However, the Kv chimeras suggest that it modulates only VSDII and VSDIII. How does one reconcile these findings? Is it that Pn3a interaction with VSDI and VSDIV does not impede movement of these VSD? Or is it that these are only weak interactions that are not relevant physiologically?

2. Several residues in CaV3.3 are proposed to mediate selective inhibition by Pn3a, however these are all based on MD simulations. It would considerably strengthen the study to provide complementary electrophysiological data either disrupting these sites (or enabling them on CaV3.1/3.2) to confirm the role of these sites experimentally.

3. One of the important advances here is that Pn3a is shown to be subtype selective for CaV3.3. Nonetheless, this toxin also blocks various CaV1/2 channels and NaV1.7. The latter is particularly high affinity (nM range) compared to CaV3.3 (uM). This limitation is noted briefly in lines 314-316. It may be helpful to expand this further for the general audience who may be interested in using this toxin practically.

*Reviewer #3 (Recommendations for the authors):*

Abstract: I am not sure that the authors are correct in saying that we do not know much about the physiological roles of Cav3.3 – there are channelopathies associated with this isoform, and there is a KO mouse that has yielded some very interesting data. Indeed, the authors acknowledge these studies in the introduction. Please tone down this statement in the abstract. No sure I agree that the S3-4 region necessarily has therapeutic potential – it will be difficult to target this site with small organic molecules – maybe stick with the molecular tool aspect instead.

Introduction- more information about the animal species of the various toxins that are discussed should be given.

Line 90 – introduction – maybe also include grammotoxin (Bean lab) here, and SNX482 (Bourinet lab) here as gating modifiers.

Line 137: Instead of referring to faster tau values, please use the term "time constants" (this also applies to others spots in the manuscript).

Line 156: At several points in the manuscript the authors use terms such as "Cav3.3's gating" – I think it would read better if this stated the "gating of Cav3.3".

Figure 2 – can you please also include IV curves rather than only the activation curves – it would allow the authors to reinforce the observed shift in the voltage dependence of activation.

Figure 2 – while the authors may be correct in assuming that there is faster recovery from inactivation with the toxin, these data could potentially be contaminated in the early phase by voltage-dependent unbinding of the toxin in response to the depolarizing pulses that are given – please recall the classical agaIVA recovery experiment from Bruce Bean's lab and also seen with SNX. It is entirely possible that the observed recovery appears accelerated because the channel not only recovers from inactivation, but also recovers from block as a result of a train of depolarizations (consistent with Figure 3).

Figure 4 title – there must be a missing word.

Figure 5 – Major point: although these data are consistent with an effect of DII VSD, one thing to keep in mind and point out clearly is that the K channel has four fold symmetry, and in these chimeras you essentially now have four Cav3.3 toxin target sites – and yet, the effect of the toxin on voltage dependence of activation of this chimera is about the same as what is observed in the Cav3.3 WT channels, suggesting that things are a bit more complicated than the reader is being led to believe. Furthermore, the chimera has completely screwed up voltage dependence of activation (i.e., 100 mV depolarized) which is problematic as this indicates that this chimera has some major structural problems. This also applies to all of the other chimeras. I think that the authors need to consider making at least one reverse chimera of Cav3.3 where Domain II VSD is replaced by either sequence from Kv1.7 or better yet, Cav3.2 or Cav3.1. Indeed, chimeras between Cav3.1 and Cav3.3 where entire membrane domains were swapped are in the public domain and may be available to the authors. I think data on a GIGG and IGII chimera pair would really add to the story. Overall, I am a little bit puzzled as to why a K channel was used fro this when the authors clearly see a large Cav3 isoform dependence and their docking model indicates that Cav3.2 can't interact the same way as Cav3.3.

Figure 6 – why is the toxin docked to all four voltage sensing domains of Cav3.3 – are the authors saying that the toxin binds to four sites simultaneously, but affects gating by only acting on DII?

In conclusion, it is a nice study, but the potassium channel backbone that is used has potential issues, not only because of the four fold symmetry, but also because of the mightily screwed up gating – you should really at least try to do a minimal set of experiments with a Cav3.1/Cav3.2-Cav3.3 chimera.

[Editors' note: further revisions were suggested prior to acceptance, as described below.]

Thank you for resubmitting your work entitled "µ-Theraphotoxin-Pn3a inhibition of Cav3.3 channels reveals a novel isoform-selective drug binding site" for further consideration by *eLife*. Your revised article has been evaluated by Richard Aldrich (Senior Editor) and a Reviewing Editor.

The manuscript has been improved but there are some remaining issues that need to be addressed, as outlined below:

Reviewer 2 has raised pertinent questions regarding complexities in correlating the electrophysiological experiments with MD simulations as it relates to the contributions of residues D689A and Q687A that can be addressed by discussion.

*Reviewer #2 (Recommendations for the authors):*

The revised manuscript has several new experiments that have strengthened the study – in particular, the chimeric experiments with CaV3.1/3.2 and CaV3.3 are insightful in deducing that DII plays a key role in binding the toxin and that this is the reason for selectivity. The mutagenesis studies are also helpful and have improved the study as a whole.

One complexity is that the correlation of electrophysiological experiments with MD simulations seems to be not as clear-cut. For instance, D689 is an important contributor in electrophysiology recordings, yet the only contact with Pn3a seems to be through backbone carbonyl which would be seemingly preserved with an alanine mutation. So, it is not fully clear why D689A has an effect while another mutation Q687A where the Q interacts with a charged residue seems to have minimal effect. Perhaps part of this is because the CaV3.1 VSDII is presumably in the up-state while Pn3a affinity is stronger in the down-state. And the orientation/accessibility of these residues may be somewhat different in the up versus down-state. It may be helpful to explicitly note what state VSD II is in for the simulations and also to briefly discuss some of these caveats.

*Reviewer #3 (Recommendations for the authors):*

Great job on the revisions. I have no further suggestions.

---

## [Author Response]

Essential revisions:1) There are some complexities associated with interpretation of the data obtained using Kv1.7/CaV3.3 VSD chimeras. Chimeras between Cav3.1 and Cav3.3 where entire membrane domains were swapped are in the public domain. CaV3.1/CaV3.3 chimeras with VSDII swapped (or similar chimera) should be done to strengthen the idea that Pn3a inhibits CaV3.3 by binding to VSDII.

A new set of experiments was performed with Cav3 domain II swap constructs including Cav3.3/Cav3.1^DII^, Cav3.3/Cav3.2^DII^, Cav3.1/Cav3.3^DII^, and Cav3.2/Cav3.3^DII^.

Consistent with the observations made in the Kv1.7/Cav3.3^DII^ system, data obtained from Cav3 domain II swap constructs supported the importance of Cav3.3^DII^ for Pn3a’s selective inhibition of Cav3.3. Grafting Cav3.3^DII^ onto Cav3.1 or Cav3.2 results in Pn3a-sensitive channels, whilst inhibition is significantly decreased when Cav3.1^DII^ (p < 0.0001) or Cav3.2^DII^ (p < 0.0001) are inserted into the Cav3.3 backbone. This data is now summarised in Figure 6E.

2) Several residues in CaV3.3 are proposed to mediate selective inhibition by Pn3a based on MD simulations. These predictions should be validated by mutagenesis and functional experiments.

The predictions of the computational model of Pn3a/Cav3.3 were examined through 7 different point mutations made in Cav3.3 (Figure 6E), and three of those were further verified in the Kv1.7/Cav3.3^DII^ system (Figure 5—figure supplement 1). From these new data sets, it can be surmised that D689 located in Cav3.3^DI^ is a key determinant of Pn3a’s affinity towards Cav3.3 with a modest contribution from E628.

3) Pn3a blocks various CaV1/2 channels and NaV1.7. The latter is particularly high affinity (nM range) compared to CaV3.3 (uM). This limitation is noted briefly in lines 314-316. The authors should expand this further for the general audience who may be interested in using this toxin practically.

We have now included additional details on Pn3a targeting of Nav and Cav channels (lines 314-319).

Reviewer #1 (Recommendations for the authors):1. It seems surprising that the scaled Pn3a-inhibited CaV3.3 currents displayed the same activation kinetics but slower inactivation compared to uninhibited channels (Figure 1). Intuitively, the inhibitory effect of Pn3a would appear to be more consistent with slower activation and faster inactivation kinetics. Single channel experiments would be needed to distinguish these two different interpretations which could both give rise to similar macroscopic currents.

We calculated the macroscopic activation and inactivation time constants from the inhibition of Cav3.3 at -20 mV by 3 µM Pn3a (Lines 126-128). There was a modest slowing of inactivation: τ_inact_ control = 39.91 ± 8.25 ms vs τ_inact_, Pn3a = 60.25 ± 16.02 ms; P = 0.04, n = 5, paired t-test. The small ~+10mV shift in Cav3.3 activation would have a modest effect on the time constant of activation and we may not be able to resolve this slowing (τ_act_, control = 5.95 ± 0.54 ms vs τ_act_, Pn3a = 6.37 ± 0.44 ms; P = 0.32, n = 5, paired t-test). The inactivation of Cav3.3 is highly dependent on Ca^2+^ influx and the Pn3a effect on recovery from inactivation may lead to the observed modest slowing of macroscopic inactivation.

2. The effects of Pn3a on CaV3.3 gating should be modeled to provide deepened insights into the possible mechanisms of action that can explain the data.

We appreciate Reviewer #1’s insightful suggestion and we will take it into account for follow-up studies.

3. Does Pn3a inhibit the maximal CaV3.3 current amplitude? The use of normalized G-V curves obscures this important point. The impact of Pn3a on Gmax should be provided.

Pn3a inhibits Cav3.3’s maximal current amplitude as shown in Figures 1A and 2A. As suggested, we have included dashed lines in Figure 2C to indicate the relative inhibitory effects of Pn3a

4. The computational docking model (Figure 6) should be validated by mutagenesis experiments.

We have validated the computational docking model through seven alanine point changes in Cav3.3 mutants and three chimeric constructs. Results of these new experiments are now presented in Figures 6E and Figure 5—figure supplement 1.

5. A significant rationale for the paper was the idea that Pn3a could potentially be used to distinguish CaV3.3-mediated effects from those of co-expressed CaV3.3 and CaV3.2 in native tissues. This capability certainly would certainly be a significant achievement. This prediction would need to be tested in a native cell type expressing multiple CaV3 channel isoforms. However, previous reports that Pn3a also inhibits Nav1.7 and HVA Ca channels suggest that interpreting the effects of the toxin in native cells is likely to be highly complicated.

Although beyond the scope of this research, follow-up studies in native tissues will ascertain the practical utility of Pn3a as a tool for the study of T-type currents mediated by Cav3.3. Much like for many other peptides and small molecules commonly used as molecular tools in the analysis of native currents, selectivity is seldom perfect (ie SNX482, 4AP, etc) and therefore the optimal experimental conditions enabling the exploitation of Pn3a’s selectivity for Cav3.3 would require optimization. These would need to take into account the specific tissue/cells in which the experiments would be performed, adequate pulse protocols (ie to remove HVA calcium channel contribution), as well as incorporating other known inhibitors such as TTX (to effectively block most Navs).

Reviewer #2 (Recommendations for the authors):The electrophysiological analysis of CaV3.x comparing the effect of Pn3a is well done and convincing. As noted in the public review, the main weakness is that the mechanism by why Pn3a accomplishes this selectivity is not fully clear and the identification of specific residues that support Pn3a interaction relies solely on computational docking.1. The mechanisms that confer subtype specificity of Pn3a for CaV3.3 is not clear. The docking simulation suggests that Pn3a interacts with both CaV3.2 and CaV3.3 at slightly distinct residues on the same, but only CaV3.3 is inhibited. Is there any experimental evidence that shows that Pn3a binds to CaV3.1/CaV3.2 VSDs? Also the docking seems to suggest that Pn3a interacts with all 4 VSDs of CaV3.3. However, the Kv chimeras suggest that it modulates only VSDII and VSDIII. How does one reconcile these findings? Is it that Pn3a interaction with VSDI and VSDIV does not impede movement of these VSD? Or is it that these are only weak interactions that are not relevant physiologically?

We apologize for the confusion caused by the early version of Figure 6A, in which Pn3a appears docked onto all four VSDs as was evaluated computationally. From the docking calculations, it was observed that the binding of Pn3a to Cav3.3^DII^ was the most energetically favourable (Figure 6—figure supplement 3) interaction observed. Consistently, electrophysiological assessment of DII swapped Cav3 constructs and Kv1.7/Cav3.3 chimeras verified that the region that enables Pn3a to inhibit Cav3.3 specifically is Cav3.3^DII^ and therefore Figure 6A was revised to reflect these results and enhance clarity.

2. Several residues in CaV3.3 are proposed to mediate selective inhibition by Pn3a, however these are all based on MD simulations. It would considerably strengthen the study to provide complementary electrophysiological data either disrupting these sites (or enabling them on CaV3.1/3.2) to confirm the role of these sites experimentally.

We appreciate the suggestion and have accordingly generated point mutations in Cav3.3^DII^ based on the MD docking calculations. These results revealed that Cav3.3DII residue D689 is important for Cav3.3/Pn3a interaction, and to a lesser extent E628, as suggested by reduced Pn3a’s inhibition observed for these mutants (Figure 6E).

3. One of the important advances here is that Pn3a is shown to be subtype selective for CaV3.3. Nonetheless, this toxin also blocks various CaV1/2 channels and NaV1.7. The latter is particularly high affinity (nM range) compared to CaV3.3 (uM). This limitation is noted briefly in lines 314-316. It may be helpful to expand this further for the general audience who may be interested in using this toxin practically.

As above, we have expanded this section to include the affinity and selectivity currently known for Pn3a on both Nav and Cav channels and as discussed further in Lines 452-460.

Reviewer #3 (Recommendations for the authors):Abstract: I am not sure that the authors are correct in saying that we do not know much about the physiological roles of Cav3.3 – there are channelopathies associated with this isoform, and there is a KO mouse that has yielded some very interesting data. Indeed, the authors acknowledge these studies in the introduction. Please tone down this statement in the abstract. No sure I agree that the S3-4 region necessarily has therapeutic potential – it will be difficult to target this site with small organic molecules – maybe stick with the molecular tool aspect instead.

We appreciate this constructive criticism and have revised the abstract and general message accordingly.

Introduction- more information about the animal species of the various toxins that are discussed should be given

We have now included scientific names for all the species addressed in the introduction.

Line 90 – introduction – maybe also include grammotoxin (Bean lab) here, and SNX482 (Bourinet lab) here as gating modifiers

We have added grammotoxin and SNX482 with appropriate references to line 90.

Line 137: Instead of referring to faster tau values, please use the term "time constants" (this also applies to others spots in the manuscript)

Corrected Lines 137 and 190.

Line 156: At several points in the manuscript the authors use terms such as "Cav3.3's gating" – I think it would read better if this stated the "gating of Cav3.3".

Corrected.

Figure 2 – can you please also include IV curves rather than only the activation curves – it would allow the authors to reinforce the observed shift in the voltage dependence of activation

We have now included the I-V curves in Figure 2A-1

Figure 2 – while the authors may be correct in assuming that there is faster recovery from inactivation with the toxin, these data could potentially be contaminated in the early phase by voltage-dependent unbinding of the toxin in response to the depolarizing pulses that are given – please recall the classical agaIVA recovery experiment from Bruce Bean's lab and also seen with SNX. It is entirely possible that the observed recovery appears accelerated because the channel not only recovers from inactivation, but also recovers from block as a result of a train of depolarizations (consistent with Figure 3)

While the depolarization required for both AgaIVA and SNX482 recovery was fairly large, the observed voltage dependence of Pn3a inhibition may contribute to the observed increased recovery from inactivation. We have now added this caveat in the discussion (lines 377-382).

Figure 4 title – there must be a missing word

Updated to read “Figure 4. Chimeric constructs of the K_V_1.7 channel with the Ca_V_3.3 voltage sensor paddles.”

Figure 5 – Major point: although these data are consistent with an effect of DII VSD, one thing to keep in mind and point out clearly is that the K channel has four fold symmetry, and in these chimeras you essentially now have four Cav3.3 toxin target sites – and yet, the effect of the toxin on voltage dependence of activation of this chimera is about the same as what is observed in the Cav3.3 WT channels, suggesting that things are a bit more complicated than the reader is being led to believe. Furthermore, the chimera has completely screwed up voltage dependence of activation (i.e., 100 mV depolarized) which is problematic as this indicates that this chimera has some major structural problems. This also applies to all of the other chimeras. I think that the authors need to consider making at least one reverse chimera of Cav3.3 where Domain II VSD is replaced by either sequence from Kv1.7 or better yet, Cav3.2 or Cav3.1. Indeed, chimeras between Cav3.1 and Cav3.3 where entire membrane domains were swapped are in the public domain and may be available to the authors. I think data on a GIGG and IGII chimera pair would really add to the story. Overall, I am a little bit puzzled as to why a K channel was used fro this when the authors clearly see a large Cav3 isoform dependence and their docking model indicates that Cav3.2 can't interact the same way as Cav3.3

We have now included data on the inhibition by Pn3a on the Cav3.3/Cav3.1^DII^ (IGII), Cav3.1/Cav3.3^DII^ (GIGG) as well as Cav3.3/Cav3.2^DII^ (IHII) and Cav3.2/Cav3.3^DII^ (HIHH). This new data highlights that Cav3.3^DII^ is critical for specific Pn3a interactions with Cav3.3.

Figure 6 – why is the toxin docked to all four voltage sensing domains of Cav3.3 – are the authors saying that the toxin binds to four sites simultaneously, but affects gating by only acting on DII?

Thank you for pointing this out. We initially modelled Pn3a interactions with all four domains of Cav3.3, to show that Pn3a preferentially targeted DII. We have updated Figure 6A to include only the most energetically favoured interaction site on Cav3.3^DII^.

In conclusion, it is a nice study, but the potassium channel backbone that is used has potential issues, not only because of the four fold symmetry, but also because of the mightily screwed up gating – you should really at least try to do a minimal set of experiments with a Cav3.1/Cav3.2-Cav3.3 chimera

[Editors' note: further revisions were suggested prior to acceptance, as described below.]

Reviewer #2 (Recommendations for the authors):The revised manuscript has several new experiments that have strengthened the study – in particular, the chimeric experiments with CaV3.1/3.2 and CaV3.3 are insightful in deducing that DII plays a key role in binding the toxin and that this is the reason for selectivity. The mutagenesis studies are also helpful and have improved the study as a whole.One complexity is that the correlation of electrophysiological experiments with MD simulations seems to be not as clear-cut. For instance, D689 is an important contributor in electrophysiology recordings, yet the only contact with Pn3a seems to be through backbone carbonyl which would be seemingly preserved with an alanine mutation. So, it is not fully clear why D689A has an effect while another mutation Q687A where the Q interacts with a charged residue seems to have minimal effect. Perhaps part of this is because the CaV3.1 VSDII is presumably in the up-state while Pn3a affinity is stronger in the down-state. And the orientation/accessibility of these residues may be somewhat different in the up versus down-state. It may be helpful to explicitly note what state VSD II is in for the simulations and also to briefly discuss some of these caveats.

We agree with Reviewer #2 in the limitations of reconciling functional observations of dynamic proteins (such as voltage-gated ion channels) and computational predictions made from homology models of, comparatively static, structures. The Cryo-EM structure of human Cav3.3 Cryo-EM became available during the revisions of our manuscript (Nat Commun. 2022 Apr 19;13(1):2084). Structural alignment revealed that the Cav3.3 (7WLI) and Cav3.1 (6KZO) voltage sensor domains appear to be in similar configurations with an RMSD (for Cα-pairs of VSD, SF, and ECLs) of 1.4 Å (He et al., 2022).

We performed new computational modelling using Cav3.3 Cryo-EM structure to assess the interactions with Pn3a. This new modelling data is consistent with the establishment of key interactions between Pn3a’s residue K17 and Cav3.3DII’s D689 (potentially through the formation of a salt bridge) as well as D689, thereby reconciling our mutagenesis results and modelling predictions. The updated Cav3.3-Pn3a binding model is now reflected in the revised Figure 6 and its supporting figures, and accordingly described/discussed in methods, results and discussion.